# Lipid Production from Sugarcane Top Hydrolysate and Crude Glycerol with *Rhodosporidiobolus fluvialis* Using a Two-Stage Batch-Cultivation Strategy with Separate Optimization of Each Stage

**DOI:** 10.3390/microorganisms8030453

**Published:** 2020-03-23

**Authors:** Jeerapan Boonyarit, Pirapan Polburee, Bongkot Khaenda, Zongbao K. Zhao, Savitree Limtong

**Affiliations:** 1Department of Microbiology, Faculty of Science, Kasetsart University, Bangkok 10900, Thailand; jeerapan.b@ku.th (J.B.); khaenda.b@gmail.com (B.K.); 2Rattanakosin College for Sustainable Energy and Environment (RCSEE), Rajamangala University of Technology Rattanakosin, Nakhon Pathom 73170, Thailand; pirapan@g.swu.ac.th; 3Department of Microbiology, Faculty of Science, Srinakharinwirot University, Bangkok 10110, Thailand; 4Division of Biotechnology, Dalian Institute of Chemical Physics, CAS, Dalian 116023, China; zhaozb@dicp.ac.cn; 5Academy of Science, Royal Society of Thailand, Bangkok 10300, Thailand

**Keywords:** microbial lipid, oleaginous yeast, two-stage cultivation, sugarcane top, crude glycerol

## Abstract

Lipids from oleaginous microorganisms, including oleaginous yeasts, are recognized as feedstock for biodiesel production. A production process development of these organisms is necessary to bring lipid feedstock production up to the industrial scale. This study aimed to enhance lipid production of low-cost substrates, namely sugarcane top and biodiesel-derived crude glycerol, by using a two-stage cultivation process with *Rhodosporidiobolus fluvialis* DMKU-SP314. In the first stage, sugarcane top hydrolysate was used for cell propagation, and in the second stage, cells were suspended in a crude glycerol solution for lipid production. Optimization for high cell mass production in the first stage, and for high lipid production in the second stage, were performed separately using a one-factor-at-a-time methodology together with response surface methodology. Under optimum conditions in the first stage (sugarcane top hydrolysate broth containing; 43.18 g/L total reducing sugars, 2.58 g/L soy bean powder, 0.94 g/L (NH_4_)_2_SO_4_, 0.39 g/L KH_2_PO_4_ and 2.5 g/L MgSO_4_ 7H_2_O, pH 6, 200 rpm, 28 °C and 48 h) and second stage (81.54 g/L crude glycerol, pH 5, 180 rpm, 27 °C and 196 h), a high lipid concentration of 15.85 g/L, a high cell mass of 21.07 g/L and a high lipid content of 73.04% dry cell mass were obtained.

## 1. Introduction

Biodiesel has largely been accepted as an alternative energy source to fossil fuel, as it is renewable and less toxic. The combustion of biodiesel is very similar to fossil-fuel combustion, but it produces fewer harmful gases, such as sulfur oxide, sulfur dioxide, and sulfur trioxide [1] which can reduce air pollution from mass transportation. Currently, biodiesel is being used worldwide as the main source of fuel for diesel engine machinery without the need for major modifications to the mechanics of these machines [2]. Biodiesel can be divided into three generations based on the feedstock which generates the fuel. First-generation biodiesel is produced from edible plant oils, such as palm oil, soybean oil, and coconut oil, and second-generation biodiesel is produced from nonedible plant oils, such as jatropha, animal fats and waste oils [3]. The most recent generation of biodiesel is derived from microbial lipids. Using recovered animal fats and frying oils of the second generation as feedstock for biodiesel can efficiently reduce the price of the fuel; however, the amount of these fats and oils is limited on an industrial scale and cannot meet the increasing needs of biodiesel production [4]. Recently, microbial lipids that are produced by oleaginous microorganisms have been getting increased attention as an oil feedstock [5]. In terms of obtainability and sustainability, lipids from oleaginous microorganisms are recognized as opportunity feedstocks for biodiesel production and provide a promising route to food and energy security in the current energy crisis [6].

Oleaginous yeasts are a yeast species that can accumulate more than 20% of their biomass in lipids during lipid production. Strains of this species include *Cryptococcus albidus, Lipomyces starkeyi, Rhodosporidium toruloides, Rhodotorula glutinis,* and *Yarrowia lipolytica* [7], and are all considered suitable microorganisms for lipid production because of their ability to use a wide range of low-cost substrates. These substrates are able to promote yeast growth to accumulate considerable quantities of lipids [8]; however, further development of the production process is necessary to bring these lipid feedstocks up to the industrial scale. A reduction in the production costs by using robust yeast strains, low-cost substrates, and optimizing the yeast cultures used during production are topics of great interest in this effort [9]. 

An advantage of the lipid production of oleaginous yeasts is that the growth rate of the yeast decelerates when the nitrogen source is exhausted. Then, the carbon source is channeled toward lipid synthesis leading to the accumulation of triacylglycerols (TAGs) within intracellular lipid bodies [10]. Many studies have reported that oleaginous yeasts synthesize and store greater quantities of TAGs when cultured in a nitrogen-limited medium containing an excess of carbon substrates [9,10,11,12,13]. The limited nitrogen condition blocks cell division and initiates lipid accumulation in oleaginous yeasts [14]. It has been reported that a high cell mass was not obtained when the oleaginous yeast was cultivated in an extremely high C/N ratio condition [15]. This demonstrated that consideration of only the initial high C/N ratio is not always sufficient for high lipid production. Various cultivation processes have been reported for lipid production in yeast; i.e., batch [16,17,18], fed-batch [19,20,21] and two-stage cultivations [22,23]. In two-stage cultivation strategies, cell division and lipid accumulation modes are spatially and/or temporally separated to achieve a high cell mass and high lipid productivity. In the first stage, cells are cultivated in a nutrient-rich medium for propagation, and in the second stage, cells are re-suspended in a limited medium containing little to no nitrogen sources, and an excess of carbon without auxiliary nutrients for the production of lipids [24,25]. Moreover, a high cell density can be obtained when yeast is cultivated in a nitrogen-rich medium with a low carbon concentration, which has shown high lipid yields as well [26].

Recent studies have focused on the use of inexpensive materials such as agricultural residues and industrial byproducts as raw materials for lipid production. Various oleaginous yeast strains haven shown the ability to use sugars derived from a wide range of lignocellulosic biomass carbon sources, which aid in sample growth and lipid accumulation [27]. Sugarcane top (ST) is an agricultural lignocellulosic waste which is annually produced in large quantities because sugarcane is one of the most widely grown crops around the world. After harvesting, the ST remains in the field where it is sometimes destroyed through burning, thus heavily polluting the surrounding environment [28]. This qualifies ST as an appropriate feedstock, due to it is abundance and ready availability. The composition of dry ST includes 42.7% cellulose, 25.5% hemicellulose and 5.8% lignin [29]. However, pretreatment and hydrolysis of lignocellulosic biomasses usually produce inhibitory compounds, such as acetic acid, furfural, and 5-hydroxymethylfurfural, formic acid, and vanillin, which could have negative effects on growth, metabolism, and product formation of microorganisms [30]. Crude glycerol (CG), a byproduct from biodiesel production plants which has been shown to have some inhibitory compounds to microorganism growth, is currently being explored as a possible large-scale carbon source in lipid production by many researchers. Many studies have demonstrated that various oleaginous yeasts have a great capacity to convert CG into lipids with high yields [23,31,32,33,34]. 

In this present study we are testing the potential of *Rhodosporidiobolus fluvialis* DMKU-SP314, which is reported to possess the ability to produce high lipid yields from lignocellulosic hydrolysate, and to grow in the presence of inhibitory compounds from sugarcane top hydrolysate (STH) [29]. In a recent study, *R. fluvialis* DMKU-SP314 was used for lipid production from STH and CG in a two-stage fed-batch cultivation processes [21]. The results of this study found that STH and CG can be used as carbon sources for growth and lipid production, respectively. Therefore, in this current study, we decided to test if enhancing lipid production from low-cost substrates in a two-stage cultivation process using *R. fluvialis* DMKU-SP314 was possible. In the first stage, cell mass was produced from STH; in the second stage these cells produced lipids from CG. Optimization for high cell mass production in the first stage for lipid production in the second stage was carried out separately by one-factor-at-a-time technique together with response surface methodology (RSM).

## 2. Materials and Methods

### 2.1. Yeast Strain and Inoculum Preparation

The *R. fluvialis* DMKU-SP314 used in this study was reported to accumulate a high lipid content (55% of dry cell mass) in nitrogen-limited medium II containing a 70 g/L mixture of glucose and xylose in a ratio of 2:1 by shaking flask cultivation at 150 rpm and 28 °C for 240 h, and, in an optimized lipid production medium for shaking flask cultivation containing STH supplemented with 59 g/L CG, 66.6% of dry cell mass was obtained [29]. The strain is still maintained on slants of yeast extract-malt extract (YM) agar (3 g/L malt extract, 3 g/L yeast extract, 5 g/L peptone, 10 g/L glucose and 20 g/L agar) at 8 °C.

The inoculum was prepared by transferring one loopful of a 48 h culture grown on a YM agar slant to 50 mL YM broth in a 250 mL Erlenmeyer flask and incubating the culture on a rotary shaker (Lab Companion IS-971R, Seoul, Korea) at 150 rpm and 28 °C for 24 h. A pre-calculated volume of inoculum was transferred to the STH medium to give an initial cell concentration determined by optical density (OD) at 600 nm of 1.

### 2.2. Substrates Used for Lipid Production

For preparation of STH, ST harvested from the field was washed with tap water to remove dirt and then dried under the sun for 2 days. Dried ST was milled and passed through a 40–60 mesh sized screen (Sieve size 0.389–0.231 mm). The milled ST was pretreated by alkaline oxidation using the method of Liu, et al. [26] and hydrolyzed by Accellerase^®^ 1500 (DuPont, Itasco, IL, USA) as described by Poontawee, et al. [30]. The total reducing sugar (TRS) in the original hydrolysate was 43.18 g/L and, to raise the TRS to a higher value, D-glucose was added to the hydrolysate.

The CG used in this study was a byproduct from the biodiesel production plant, Thai Oleochemicals Co. Ltd., Bangkok, Thailand. It contained 82.06% glycerol, 0.10% methanol, 3.52% ash, 5.43% non-glycerol organic matter and 8.89% water [23]. The CG was used as the sole carbon source in the second stage of cultivation.

Soybean powder (SBP) used in this study was purchased from Doi Kham Food Products Co. Ltd., Bangkok, Thailand and consisted of 38.7% protein that was calculated to have a total nitrogen content of 7%, 5% fat, 1% carbohydrate, 8% iron and 6% calcium.

### 2.3. Lipid Production from STH and CG by Two-Stage Batch Cultivation

In this study, a two-stage cultivation process was used. In the first stage, optimization of cell mass production from STH was performed. In the second stage, optimization for lipid production from CG was carried out. Cells at the end of the first stage of cultivation were collected by centrifugation and re-suspended in a CG solution in the second stage.

#### 2.3.1. Optimization of Cell Mass Production from STH in the First Stage

##### Effect of Additional Nitrogen Source

Various nitrogen compounds consisting of inorganic nitrogen compounds, such ammonium sulfate [(NH_4_)_2_SO_4_] and ammonium chloride (NH_4_Cl), and organic nitrogen compounds, such as corn steep liquor (CSL), monosodium glutamate (C_5_H_8_NaO_4_), soybean powder (SBP), and urea [CO(NH_2_)_2_], were used as additional nitrogen sources in comparison with the yeast extract (YE). Cultivation was performed in a 500 mL Erlenmeyer flask which contained 100 mL of the STH medium, the composition of which was modified from the nitrogen-limited medium II [35]. The STH medium was composed of STH giving 40 g/L TRS, 0.55 g/L (NH_4_)_2_SO_4_, 0.4 g/L KH_2_PO_4_, 2.0 g/L MgSO_4_ 7H_2_O and an individual additional nitrogen source at the equivalent nitrogen content of 0.6 g/L (C/N ratio was 6.67) with an initial pH of 5.5.

In the second stage, cells from the first stage were harvested by centrifugation at 8000 rpm for 5 min and re-suspended in 70 g/L of CG solution adjusted to pH 5.5. The incubation was performed on a rotary shaker (Lab Companion IS-971 R, Seoul, Korea) at 150 rpm and 28 °C for 120 h of each stage. In both stages, every 24 h cells were harvested for analysis of cell mass and lipid concentration. The additional nitrogen source that provided a high cell mass concentration with the highest lipid accumulation was selected for further studies.

##### Screening of Significant Factors for Cell Mass

In this study, eight factors consisting of medium component parameters (X_1_; concentration of TRS, X_2_; additional nitrogen sources, X_3_; (NH_4_)_2_SO_4_, X_4_; KH_2_PO_4_, and X_5_; MgSO_4_ 7H_2_O) and physical parameters (X_6_; initial pH, X_7_; temperature and X_8_; shaking speed) were selected. The Plackett–Burman design (PBD) was used to screen important medium components and conditions with respect to their effects on cell mass production. The significant factors for cell mass and lipid production with levels of confidence higher than 90%, and the levels that gave the highest concentration of cell mass and lipid yield, were selected for further studies.

##### Optimization Temperature by the One-Factor-At-a-Time Methodology

Optimal temperature for biomass production was determined by the one-factor-at-a-time methodology experiment using a temperature gradient incubator (Toyo Kagaku Sangyo Co. Ltd., Tokyo, Japan). The temperature was set between 15.0 and 35.5 °C and an L-shaped tube containing 10 mL STH medium with the selected additional nitrogen source was used. Cultivation was performed in duplicates. After 120 h, cells were harvested for analysis of cell mass and lipid concentration. The optimal temperature for growth in hydrolysate medium was selected for further studies.

##### Optimization of the Concentration of (NH_4_)_2_SO_4_ and KH_2_PO_4_, and Shaking Speed by Box-Behnken Design

The Box-Behnken design (BBD) was applied to further identify the optimal medium components and conditions for cell mass production cultivated in the STH medium during the first stage of the two-stage cultivation process. Three factors, the concentration of (NH_4_)_2_SO_4_, of KH_2_PO_4_, and the shaking speed, were used to determine the response pattern and synergy of the factors. The 17 experimental runs of the BBD matrix were conducted in 500 mL Erlenmeyer flasks containing 100 mL of the STH medium with different concentrations of (NH_4_)_2_SO_4_ at 0.1–1.7 g/L and KH_2_PO_4_ at 0.01–0.39 g/L. The temperature and shaking speed were controlled at 28 °C and 180–200 rpm, respectively. The average cell mass production was taken as a dependent or response factor (Y_1_). Design-Expert Software evaluated the integrity of the regression model using the coefficient of determination (*R^2^*) and the analysis of variance (ANOVA).

##### Shift Time Improvement

The effect of the different shift times on cell mass production in the first stage was studied using the optimal STH medium and conditions determined by the RSM. Cells were harvested by centrifugation after cultivation for 48, 72, 96, and 120 h and resuspension in CG solution.

##### Initial TRS Concentration

To obtain the appropriate concentration of initial TRS of the STH medium after testing the effect of shift time, the initial TRS concentration was adjusted to 40, 50, 60, 70 and 80 g/L.

#### 2.3.2. Optimization of Lipid Production from CG in the Second Stage

The optimization of lipid production from CG in the second stage was performed by the one-factor-at-a-time methodology and BBD. The optimal temperature for lipid production was determined by using a temperature gradient incubator with a temperature range between 17.0 and 37.8 °C. The cells were cultivated in 10 mL of optimal STH medium and, under optimal cultivation conditions, were collected and re-suspended in a 10 mL of 70 g/L CG solution with an initial pH of 5.5 in an L-shaped tube.

To identify the optimal conditions for the lipid concentration and lipid content of *R. fluvialis* DMKU-SP314 cultivated in the CG in the second stage of the two-stage cultivation process, a BBD was employed. Three factors, namely CG concentration, initial pH and shaking speed, were used to investigate the interactive effects of each variable on lipid production by using the experimental design. The first stage of cultivation was performed in STH with the optimal nutrients and cultivation conditions set from previous experiments. Then the cells were harvested and re-suspended in a CG solution in the second stage of cultivation. The 17 experimental runs from the BBD were conducted in 500 mL Erlenmeyer flasks containing 100 mL of different CG concentrations ranging between 70–130 g/L, with the initial pH adjusted to 5 or 6. The temperature and shaking speed were controlled at 27 °C and 180–200 rpm, respectively.

The average cell mass production, lipid concentration, and lipid content were taken as dependent or response factors (Y_1_, Y_2_, and Y_3_, respectively). The integrity of the regression model was evaluated using the coefficient of determination (*R^2^*) and the ANOVA.

### 2.4. Analytical Methods

To analyze the cell mass concentration (g/L), cells were harvested by centrifugation from a culture broth and washed twice with distilled water. The harvested cells were dried at 80 °C until they reached a constant weight of the cell mass.

To determine the lipid concentrations (g/L), lipids were extracted from the harvested cells by modification of the method of Bligh and Dyer [36]. Fatty acids were transformed into methyl ester through transesterification through the method of Holub and Skeaff [37]. The fatty acid methyl ester was analyzed using a gas chromatograph (GC14-A, Shimadzu, Kyoto, Japan) with a flame ionization detector and a silica megabore capillary column (30 m × 0.52 mm × 1 μm, Durabond 225, J and W Scientific, Texas, TX, USA). Lipid concentration was expressed as grams of the lipid (the sum of fatty acid concentration) per milliliter of the culture broth (g/L). Lipid content was expressed as the percentage of grams of lipid concentration per grams of dry cell mass (% of dry cell mass).

The TRS was measured by the dinitrosalicylic acid (DNS) method [38]. Glycerol concentration (g/L) was analyzed by using a Glycerol Colorimetric Assay Kit (Sigma-Aldrich, St. Louis, MO, USA).

## 3. Results and Discussion

To enhance lipid production from two low-cost substrates i.e., STH and CG by the selected oleaginous strain, *R. fluvialis* DMKU-SP314, a two-stage batch cultivation process was used. Separate optimizations of cell mass production in the first stage and lipid production in the second stage were performed. In the first stage, optimization of cell mass production was performed in enzymatic hydrolysis STH containing 43.18 g/L TRS. In the second stage, lipids were produced by cultivation in a CG solution.

### 3.1. Optimization of Cell Mass Production from STH in the First Stage

#### 3.1.1. Effect of Additional Nitrogen Source

To find the most appropriate nitrogen source for cell mass production, inorganic nitrogen compounds i.e., (NH_4_)_2_SO_4_ and NH_4_Cl, and organic nitrogen compounds i.e., CSL, C_5_H_8_NNaO_4_, SBP, CO(NH_2_)_2_ and YE were individually tested at the same nitrogen concentration. At the end of the first stage, the highest cell masses of 9.62, 9.60, 9.08 and 8.74 g/L were obtained in the STH medium supplemented with CO(NH_2_)_2_, CSL, SBP, and C_5_H_8_NNaO_4_, respectively (Figure 1).

Considering lipid concentration and lipid content after the second stage, cells obtained when cultivated in the STH medium supplemented with CSL and SBP were much higher than the other combinations at 8.90 g/L and 61.85%, and 8.74 g/L and 63.83%, respectively. YE provided the lowest cell mass (6.48 g/L) at the end of first stage and relatively low lipid concentration and content (2.54 g/L and 16.87%) after the second stage. Low cell mass production in the first stage and lipid production in the second stage resulted from using inorganic nitrogen compounds. The results revealed that either CSL or SBP could be used as additional nitrogen sources. According to the results, SBP provided a higher lipid content than CSL, which indicate a higher potential of yeast cells obtained for lipid accumulation. This result corresponded to the report of Poontawee, et al. [30], where lipid concentration was significantly increased when the cultivation medium was supplemented with SBP. Therefore, SBP was selected as the additional nitrogen source for further studies.

#### 3.1.2. Screening of Significant Factors

Eight factors (including the concentration of TRS, the additional nitrogen source, (NH_4_)_2_SO_4_, KH_2_PO_4_, MgSO_4_ 7H_2_O, the initial pH, temperature and shaking speed) were analyzed with regard to their effect on cell mass production at the end of the first stage by RSM using PBD. The coefficient values from the regression analysis are shown in Table 1.

Generally, factors with a higher *T*-value and lesser *p*-value were considered to be significant model terms [39]. Factors with levels of confidence higher than 90% were considered to have a significant effect on the response. The results showed that four factors, X_3_ ((NH_4_)_2_SO_4_ concentration), X_4_ (KH_2_PO_4_ concentration), X_7_ (temperature) and X_8_ (shaking speed), most affected cell mass production. The factors i.e., (NH_4_)_2_SO_4_ concentration and shaking speed showed significant positive effects, while KH_2_PO_4_ concentration and temperature showed significant negative effects. Shaking speed, with a probability value of 0.022, was found to be the factor with the highest influence on cell mass production.

#### 3.1.3. Optimization of Temperature by the One-Factor-At-a-Time Method

Temperature was found to be an important significant factor in the PBD experiment. To obtain an accurate optimal temperature of yeast growth, temperature optimization was conducted separately using a temperature gradient incubator. The statistical analysis showed the optimal temperature for growth of this strain was between 27 and 28 °C (data not shown). In this experiment, the highest cell mass was achieved when cultivation was conducted at 28 °C for 120 h.

In the previous study conducted by Poontawee, et al. [30], a temperature of 28 °C was also used with this strain, *R. fluvialis* DMKU-SP314, and the same species, *R. fluvialis* DMKU-RK253, to provide high cell mass and lipid concentrations [30]. Moreover, this temperature was also used for cultivation of other yeast species *Rhodotorula toruloides*, *R. glutinis*, *T. coremiiforme* and *Y. lipolytica* for growth and lipid production [12,35,40,41]. Therefore, 28 °C was selected for further studies.

#### 3.1.4. Optimization of the Concentration of (NH_4_)_2_SO_4_, KH_2_PO_4_, and Shaking Speed by BBD

Optimization was performed by the construction of a quadratic model using BBD. The influence of the three factors, X_1_ ((NH_4_)_2_SO_4_ concentration), X_2_ (KH_2_PO_4_ concentration) and X_3_ (shaking speed) on cell mass production with *R. fluvialis* DMKU-SP314 were determined at the optimal temperature, 28 °C. To design the three significant factors used in the present study, the middle point values of factors were set at the levels that gave maximal yield of cell mass based on the results of the PBD experiment, except for shaking speed. The shaking speed was set as 160, 180, and 200 rpm. For the insignificant factors, the levels that gave the maximal cell mass concentration from the PBD experiment were used (including 70 g/L TRS, 2.58 g/L SBP, 2.5 g/L MgSO_4_ 7H_2_O and an initial pH of 6). After an incubation period of 120 h, the maximal cell mass of 16.73 g/L was obtained. The data were analyzed by multiple regression analyses, and the regression coefficients were determined (Table 2).

The quadratic model was highly significant, with a very high model *F*-value (69.42) and a very low *p*-value (*p* < 0.0001). In addition, any lack of integrity in the *F*-value of the model was not significant (0.1636). Furthermore, the value of the correlation coefficient (*R*^2^) for cell mass was 0.9889, and the adjusted *R*^2^ was 0.9747. These values indicate that the quadratic model was significant and there was a high correlation between the actual results and the values predicted by the model equation. The ANOVA of the optimization of cell mass production indicated that among the three factors selected by the PBD experiment, shaking speed (X_4_) and (NH_4_)_2_SO_4_ concentration (X_1_) had significant effect on cell mass production, contrary to the KH_2_PO_4_ concentration (X_2_) (Table 2). Moreover, the interaction between shaking speed and the (NH_4_)_2_SO_4_ concentration (X_1_X_3_) were also significant. However, the interaction between the (NH_4_)_2_SO_4_ concentration and KH_2_PO_4_ concentration (X_1_X_2_), and the interaction between shaking speed and the KH_2_PO_4_ concentration (X_2_X_3_) were not significant, indicating that the KH_2_PO_4_ concentration had no significant influence on the response.

The BBD gave the following second-order polynomial equation for the cell mass (Y) production as a function of X_1_ [(NH_4_)_2_SO_4_ concentration], X_2_ (KH_2_PO_4_ concentration) and X_3_ (shaking speed).
(1)$$Y=+13.77+2.72X_{1}+0.15X_{2}+1.40X_{3}+0.025X_{1}X_{2}\;+1.72X_{1}X_{3}\;-0.047X_{2}X_{3}\;-2.63X_{1}^{2}+0.077X_{2}^{2}\;-0.25X_{3}^{2}$$

A significant positive effect of the (NH_4_)_2_SO_4_ concentration with respect to cell mass production was also found in *Candida* sp. and *R. glutinis* [42]. According to the ANOVA results, the (NH_4_)_2_SO_4_ concentration showed a significant negative effect on lipid concentration and lipid content (data not shown). This was also observed by Bandhu et al. [43] who found from BBD that (NH_4_)_2_SO_4_ concentration had significant negative effect on lipid concentration. Consequently, a decrease of the (NH_4_)_2_SO_4_ concentration led to enhanced lipid concentration together with lipid content.

This experiment aimed to optimize cell mass production; thus, the significant factors that provided the highest cell mass concentration were selected for validation according to the computational model. The model predicted values of 15.34 g/L cell mass, 5.11 g/L lipid concentration and 33.31% lipid content when cultivated in a hydrolysate medium composed of 0.94 g/L (NH_4_)_2_SO_4_ and 0.39 g/L KH_2_PO_4_ on a rotary shaker at 200 rpm and 28 °C. The solution was tested in a shaking flask using the values depicted as optimal, which resulted in cell mass and lipid concentrations of 15.60 g/L and 6.18 g/L, respectively, calculated at a lipid content of 40.06% of dry cell mass at the end of the first stage (120 h).

#### 3.1.5. Shift Time Improvement

In two-stage cultivation, the shift from the first stage to the second stage may influence lipid production. Therefore, we investigated the effect of shift time on lipid production. In all the previous experiments cited in this study, the first stage was carried out for 120 h and then cells were collected and re-suspended in the CG solution in the second stage. During the first stage, we observed a reduction in growth after 48 h (data not shown). To identify the most suitable cell stage for the shift to the second stage, the effect of shift time on cell growth and lipid production were studied at 48, 72, 96, and 120 h. The highest cell mass of 22.87 g/L, lipid concentration of 14.69 g/L and lipid content of 64.33% of dry cell mass were obtained at the end of the second stage (240 h) when the shift time was 48 h (Figure 2). Growth characteristics of this yeast demonstrated that at 48 h of cultivation, cells were in the exponential phase (data not shown). Thus, the most suitable cell stage for the shift to the second stage is indicated to be the exponential phase. For this result, the shift time of 48h in the two-stage cultivation was selected for further study.

#### 3.1.6. TRS Concentration

An initial reducing sugar concentration of 70 g/L was used for BBD optimization with the shift time at 120 h; however, the result of shift time experiments showed that 48 h was the most suitable, therefore, initial TRS of 70 g/L was probably excessive for cell propagation in the first stage. Thus, a study of the effect of initial TRS concentration ranging from approximately 40 g/L (only natural TRS in STH) to 80 g/L (TRS in STH supplemented with glucose) on cell mass and lipid production in the first stage was performed. The results in Table 3 revealed that increasing the initial TRS concentration between 43.18 and 80.40 g/L did not increased of TRS consumption or cell mass production, but rather an increase in lipid concentration and lipid content was shown. The result indicated that only TRS in STH (43.18 g/L) without the addition of sugar was sufficient for cell propagation for 48 h in the first stage of cultivation.

This result was in line with the results observed in *R. toruloides* Y4, which showed that cultivation in 40 g/L initial glucose resulted in the highest specific growth rate [44]. It is well known that initial carbon concentration affects the specific growth rate. The expression of a large number of genes that are required for the metabolism of alternative carbon sources were repressed when cells were grown in an inopportune glucose concentration [45].

In summary, the optimal hydrolysate medium for cell growth in the first stage consisted of original STH (43.18 g/L TRS), 2.58 g/L SBP, 0.94 g/L (NH_4_)_2_SO_4_, 0.39 g/L KH_2_PO_4_, 2.5 g/L MgSO_4_ 7H_2_O and an initial pH of 6. The optimal cultivation conditions were shaking on a rotary shaker at 200 rpm and 28 °C for 48 h. Under these optimal STH medium and cultivation conditions, cell mass reached 10.61 g/L with a lipid content of 5.66%.

### 3.2. Optimization of Lipid Production from CG in the Second Stage by the One-Factor-At-a-Time Technique and BBD

It has been reported that lipid production is influenced by cultivation temperature [46]. The effect of temperature on the lipid production of *R. fluvialis* DMKU-SP314 was determined by a temperature gradient incubator. The results showed that the highest lipid concentration was achieved after the second stage of cultivation, and when incubated at 27 °C (Figure 3). However, the lipid concentrations obtained when *R. fluvialis* DMKU-SP314 was cultivated at 24–27 °C were not statistically different from cultivation at 28–37.8 °C which decreased with the increase of temperature. Our results were similar to results reported by Zhang, et al. [22], which found that lipid production with *R. glutinis* CGMCC 2258 at 24 °C was higher than at 30 °C [46]. Oleaginous yeasts attained the highest cell mass and lipid accumulation under certain conditions, which included temperatures maintained between 25 and 30 °C [14]. 

Three important factors, namely CG concentration, initial pH, and shaking speed were used for optimization of lipid concentration and lipid content in the second stage of two-stage cultivation. When the *R. fluvialis* DMKU-SP314 was cultivated under optimum conditions in the first stage, and within various conditions during the second stage according to the BBD, 17 experimental runs were conducted. The results show cell mass concentrations ranging between 20.99–26.07 g/L, lipid concentrations between 10.84–14.16 g/L and lipid content between 41.64–57.39% of dry cell mass. The ANOVA for the response surface quadratic model for cell mass production, lipid production, and lipid content are presented below in Table 4.

Based on the ANOVA analysis that gave the level of response as a function of the four independents factors by employing multiple regression analysis, the following regression equations were obtained:(2)$$Y_{1}=743.0128\;-0.42994X_{1}\;-11.0167X_{2}\;-7.04754X_{3}+0.042333X_{1}X_{2}+0.000246X_{1}X_{3}+0.000005X_{2}X_{3}\;-0.000115X_{1}^{2}\;+0.58167X_{2}^{2}+0.017779X_{3}^{2}$$
(3)$$Y_{2}=407.0596\;-0.54304X_{1}-62.0692X_{2}-2.064X_{3}\;+0.000617X_{1}X_{2}+0.00027X_{1}X_{3}+0.324X_{2}X_{3}$$
(4)$$Y_{3}=-249.98662\;-1.09622\;X_{1}\;-187.2125X_{2}+9.27508X_{3}-0.068333X_{1}X_{2}\;+0.000616X_{1}X_{3}+1.2615X_{2}X_{3}\;-0.000114X_{1}^{2}\;-4.20333X_{2}^{2}\;-0.044333X_{3}^{2}$$

The relationships among the three factors X_1_ (CG concentration), X_2_ (initial pH), and X_3_ (shaking speed) were determined using the response surface for the quadratic model as cell mass production, lipid concentration, and lipid content. The regression models accurately described the experimental data, which indicated a correlation among the three factors which affected the three responses as discussed above. This statement is supported by the fact that the values of the correlation coefficient (*R^2^*) for cell mass production, lipid concentration, and lipid content were 0.94, 0.92, and 0.95, respectively, which in turn suggests that most of the errors/variation in the model can be explained [47]. It is known that the *R*^2^ value is always between 0 and 1 and that a value closer to 1 indicates stronger models and better predictions of the responses. The *R^2^* values in this study suggested a satisfactory representation of the process model and a good correlation between the experimental results and the theoretical values predicted by the model equation. The results of the regression coefficients revealed that cell mass production, lipid concentration, and lipid content were clearly affected by interactions between the shaking speed and CG concentration.

The significant influence of CG and shaking speed in relation to the cell mass and lipid concentration resulting in a higher lipid content was also observed in *R. fluvialis* by DMKU-RK253 [23]. Shaking speed has an effect on the glycerol concentration due to the fact that the oxygen supplied by shaking is required as the final electron acceptor for glycerol oxidation [48]. The dissolved oxygen concentration in the culture medium is correlated with the activity of glycerol use enzymes, which leads to high lipid production.

The shaking speed supplies the oxygen required for yeast growth in the culture broth, and, as a result, different speeds resulted in different levels of oxygen dissolution. As mentioned previously when discussing the PBD experiment, shaking speed was found to be the factor with the highest influence on cell mass and lipid concentration (Table 1). This relationship was also found in the BBD experiment (Table 2 and Table 4). Both the PBD and the BBD experiments indicated that shaking speed strongly influenced not only cell mass concentration, but also lipid concentration. This suggested that the cell mass and lipid production of *R. fluvialis* DMKU-SP314 required high oxygen availability. The effect of oxygen availability on cell mass production was also observed in *Rhodosporidium azoricum*, where oxygen was limited; the final cell mass was lower than when oxygen was not limited. Likewise, a negative effect on the lipid production of *R. azoricum* and *T. oleaginosus* was found when the aeration rate decreased [49].

Based on the high statistical significance of the regression, an RSM mathematical model was used to calculate the conditions under which the lipid production could be optimized. According to the analysis by Design-Expert software, five experimental designs were selected for validating the model and predicting optimal conditions (Table 4). The results showed that the optimal conditions for lipid production with *R. fluvialis* DMKU-SP314 corresponded to 87.54 g/L CG with an initial pH of 5 and a shaking speed of 180 rpm. These conditions are likely to provide the best process response, leading to a maximum lipid content of 58.04% of dry cell mass when the cell mass and lipid concentrations were 25.12 g/L and 14.66 g/L, respectively. The validation of the prediction experiment was carried out using the above optimal conditions, resulting in a lipid content of 73.04% of dry cell mass when the cell mass and lipid concentration were 21.70 g/L and 15.85 g/L, respectively.

The two-stage cultivation under the optimal nutrient and cultural conditions in this study could enhance lipid biosynthesis (Table 5).

In the first stage, when STH was used as a feedstock, there was a cell mass yield of up to 0.35 g/g TRS with only a low lipid yield of 0.02 g/g. This supported STH as a suitable carbon source for cell growth. Then, high lipid yield was observed in the second stage using a CG solution without auxiliary nutrients. A higher lipid yield of 0.23 g/g and cell mass yield of 0.17 g/g glycerol was obtained. These results therefore demonstrate that the two-stage cultivation process, in which STH was used in the first stage and CG was used in the second stage, produced optimal cell growth and lipid production. This provided the highest lipid content (73% of dry cell mass) when compared with other investigations, as shown in Table 6. Comparing this result with the same yeast strain and substrate revealed that using a two stage batch cultivation could enhance lipid content better than a two-stage fed-batch cultivation that was used in the previous study [21]. Although lipid concentration produced in this study was lower than that produced by *Cryptococcus curvatus* ATCC 20509, using a two-stage fed-batch process in a 1.5 L bioreactor cultivated in only CG [31]. However, nearly the same lipid concentration was obtained when the production was carried out by another strain of *R. fluvialis* using only CG and a two stage batch, shaking flask cultivation [23]. In addition, lipid concentration in this study was higher than those reported in many investigations also using a two-stage cultivation process with various yeast species and various carbon substrates, as shown in Table 6. This demonstrated that STH is a suitable substrate for cell mass production with *R. fluvialis* DMKU–SP314. The use of a two-stage cultivation process led to high cell density and an increase in a high quantity of lipids. Moreover, CG was found to be an effective carbon source for lipid production in the second stage of cultivation with *R. fluvialis* DMKU–SP314.

### 3.3. Fatty Acid Compositions

The fatty acid compositions of lipids produced by *R. fluvialis* DMKU-SP314 in this study was determined by gas chromatography after a transesterification step. The highest fatty acid was found to be oleic acid (C18:1), followed by palmitic acid (C16:0), linoleic acid (C18:2), stearic acid (C18:0), linolenic acid (C18:3) and myristic (C14:0). Similar fatty acid compositions were reported for *R. fluvialis* DMKU-SP314 in previous study [21], for *R. fluvialis* DMKU-RK253 grown on CG [23], and *R. toruloides* grown on corn stover hydrolysate [53]. Increased concentrations of unsaturated fatty acids are essentially required to improve the cold-flowing properties of biodiesel (FAMEs). However, they are undesirable for Green-diesel production due to elevated hydrogen consumption [54]. The fatty acid composition of lipids produced by this yeast species is qualitatively similar to that of common plant oils used in biodiesel production [55] as shown in Table 7.

## 4. Conclusions

This study demonstrated that STH is a suitable substrate for cell mass production by *R. fluvialis* DMKU–SP314. The use of a two-stage batch cultivation led to high cell density and an increase in a high quantity of lipids. Moreover, CG was found to be an effective carbon source for lipid production in the second stage of cultivation with *R. fluvialis* DMKU–SP314. The lipid volume produced by this yeast species exhibited a fatty acid composition similar to that of major plant oils commonly used in biodiesel production currently. Therefore, *R. fluvialis* DMKU–SP314 has a promising ability to convert agricultural residues and hydrolysates into microbial lipids with no additional nitrogen source needed in the lipid accumulation stage in order to produce lipid content viable as a renewable biodiesel fuel.

## Figures and Tables

**Figure 1 microorganisms-08-00453-f001:**
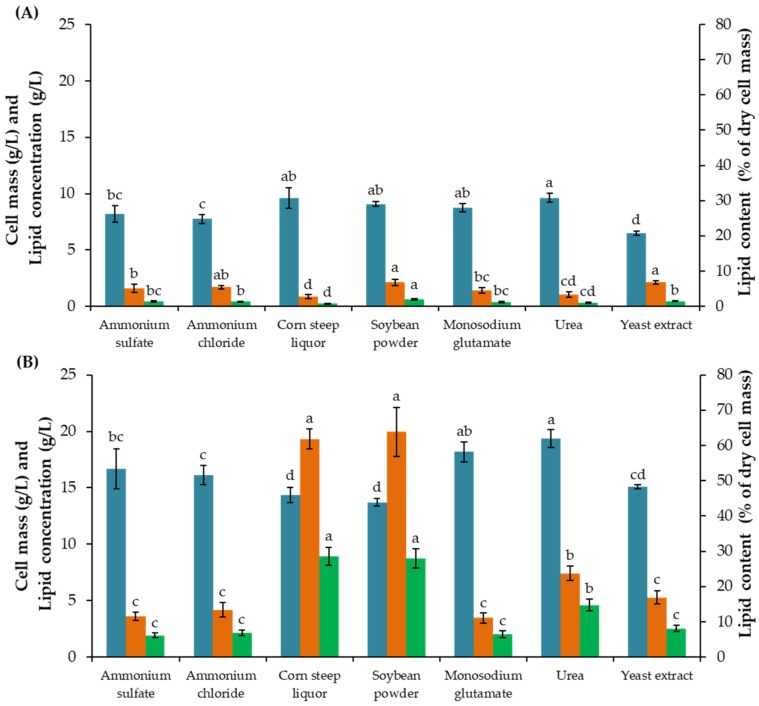
Cell mass (■), lipid content (■) and lipid concentration (■) by *R. fluvialis* DMKU-SP314 cultivated in a STH medium in various additional nitrogen compounds (**A**) at the end of the first stage (120 h) and (**B**) at the end of the second stage (240 h) in 70 g/L of CG solution with a pH of 5.5 while incubated at 150 rpm and 28 °C throughout the cultivation. Data are presented as mean value ± standard deviation. The different, same and overlapping lower-case letters mean significantly different, no-significantly different, and on-significantly different of their overlapping according to Duncan’s multiple range test at *p* ≤ 0.05.

**Figure 2 microorganisms-08-00453-f002:**
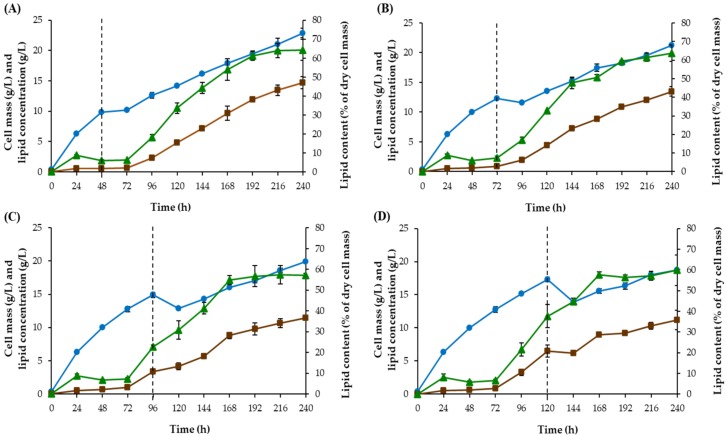
Time-course of cell mass (●), lipid content (▲) and lipid concentration (■) of *R. fluvialis* DMKU-SP314 cultivated in a STH medium consisting of 70 g/L TRS, 2.58 g/L SBP, 0.94 g/L (NH_4_)_2_SO_4_, 0.39 g/L KH_2_PO_4_, 2.5 g/L MgSO_4_ 7H_2_O, with an initial pH of 6, a temperature of 28 °C and a shaking speed of 200 rpm in the first stage. In the second stage, the cells were re-suspended in 70 g/L of a CG solution with a pH of 5.5 and incubated at 150 rpm and 28 °C at different shift times (**A**) 48 h (**B**) 72 h (**C**) 96 h and (**D**) 120 h. Data are presented as mean value ± standard deviation.

**Figure 3 microorganisms-08-00453-f003:**
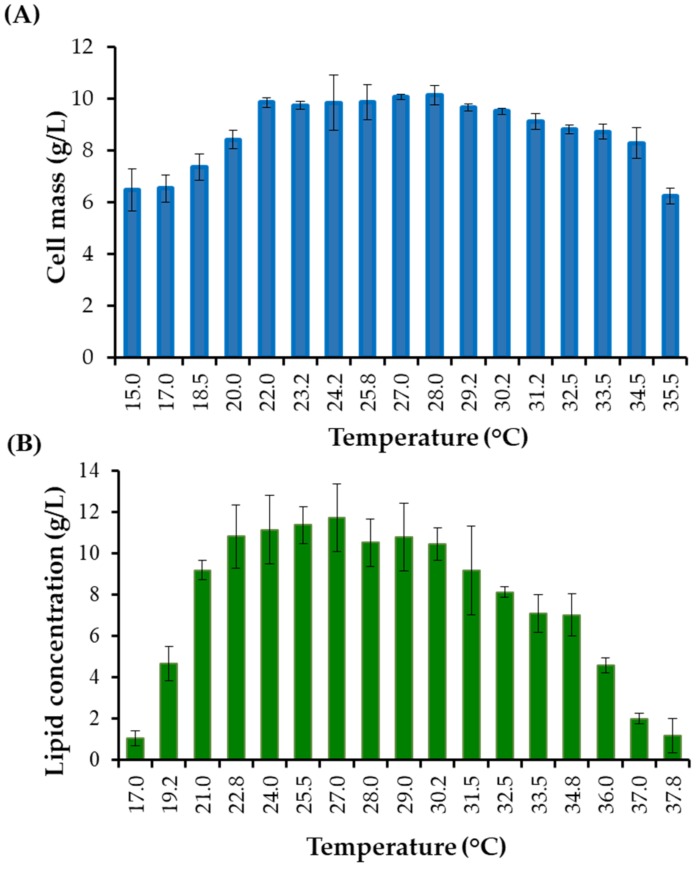
The effect of temperature on **(A)** cell mass production (g/L) when cultivated in a STH medium supplemented with SBP at 120 h in a temperature gradient incubator (15.0–35.5 °C). The effect of temperature on **(B)** lipid concentration (g/L) when cultivated in a CG solution for 196 h (after cultivation in the first stage for 48 h) in a temperature gradient incubator (17.0–37.8 °C). Data were presented as mean value ± standard deviation.

**Table 1 microorganisms-08-00453-t001:** Statistical analysis of factors using Plackett–Burman Design.

Source	Cell Mass		Lipid Concentration		Lipid Content	
	*T*-Value	*p*-Value	*T*-Value	*p*-Value	*T*-Value	*p*-Value
X_1_	1.8151	0.1671	2.0491	0.1329	1.8485	0.1617
X_2_	1.3260	0.2767	2.2091	0.1142	2.3541	0.0999
X_3_	2.7722	0.0694	−1.0801	0.3592	−4.3845	0.0220
X_4_	−2.6963	0.0740	−2.6091	0.0797	−2.6770	0.0752
X_5_	−0.1623	0.8814	−0.9734	0.4021	−1.4697	0.2380
X_6_	−0.9086	0.4305	0.3334	0.7608	0.6194	0.5795
X_7_	−2.6626	0.0762	−1.4446	0.2443	−0.6970	0.5359
X_8_	4.3660	0.0222	4.1826	0.0249	3.9307	0.0293

X_1_ = TRS, X_2_ = Soybean powder, X_3_ = (NH_4_)_2_SO_4_, X_4_ = KH_2_PO_4_, X_5_ = MgSO_4_ 7H_2_O, X_6_ = Initial pH, X_7_ = Temperature and X_8_ = Shaking speed. *R*^2^ = 0.9400 for cell mass, *R*^2^ = 0.9263 for lipid concentration, *R*^2^ = 0.9472 for lipid content.

**Table 2 microorganisms-08-00453-t002:** Analysis of variance for the quadratic response surface model on first-stage cultivation process.

Source	SS	DF	MS	Cell Mass	
				*F*-Value	*p*-Value
Model	116.85	9	12.98	69.42	<0.0001
X_1_	59.34	1	59.34	317.30	<0.0001
X_2_	0.17	1	0.17	0.93	0.3677
X_3_	15.71	1	15.71	84.00	<0.0001
X_1_X_2_	0.00	1	0.00	0.01	0.9112
X_1_X_3_	11.79	1	11.79	63.03	<0.0001
X_2_X_3_	0.01	1	0.01	0.05	0.8334
X_1_^2^	29.15	1	29.15	155.85	<0.0001
X_2_^2^	0.03	1	0.03	0.13	0.7249
X_3_^2^	0.27	1	0.27	1.44	0.2684
Lack of Fit	0.90	3	0.30	2.92	0.1636
*R* ^2^					0.9889
Adjust *R*^2^					0.9747

SS = sum of squares, DF = degree of freedom, MS = mean square, 95% significant level. X_1_ = (NH_4_)_2_SO_4_ (g/L), X_2_ = KH_2_PO_4_ (g/L) and X_3_ = Shaking speed (rpm).

**Table 3 microorganisms-08-00453-t003:** Total reducing sugar consumption, cell mass, lipid concentration, and lipid content by *R. fluvialis* DMKU-SP314 at 48 h of first-stage cultivation.

TRS (g/L)	TRS_C_ (% of Initial TRS)	Cell Mass (g/L)	Lipid Concentration (g/L)	Lipid Content (% of Dry Cell Mass)
43.18	69.54 ^a^	10.61 ± 0.22 ^a^	0.65 ± 0.09 ^b^	5.66 ± 0.71 ^c^
52.21	54.22 ^b^	10.54 ± 0.16 ^a^	0.76 ± 0.08 ^a^	7.21 ± 0.82 ^b^
62.87	45.55 ^ab^	10.46 ± 0.13 ^a^	0.81 ± 0.08 ^a^	7.70 ± 0.69 ^ab^
72.26	39.05 ^b^	9.94 ± 0.22 ^b^	0.85 ± 0.05 ^a^	8.58 ± 0.65 ^a^
80.40	34.21 ^b^	9.93 ± 0.28 ^b^	0.87 ± 0.02 ^a^	8.70 ± 0.40 ^a^

^a^ TRS = initial total reducing sugar concentration (g/L). ^b^ TRS_C_ = Percentage of total reducing sugar consumption (% of initial TRS). ^c^ Data followed by difference letters show significant difference according to Duncan’s Multiple Range Test (*p* < 0.05).

**Table 4 microorganisms-08-00453-t004:** Analysis of variance for the quadratic response surface model in the second stage cultivation process.

Source	DF	Cell Mass Production (Y_1_)		Lipid Concentration (Y_2_)		Lipid Content (Y_3_)	
		*F*-Value	*p*-Value	*F*-Value	*p*-Value	*F*-Value	*p*-Value
Model	9	8.44	0.0151	14.66	0.0006	10.98	0.0084
X_1_	1	27.08	0.0035	0.71	0.4240	13.14	0.0151
X_2_	1	1.45	0.2817	0.15	0.7071	0.24	0.6480
X_3_	1	0.60	0.474	0.76	0.4098	0.08	0.7876
X_1_ X_2_	1	3.68	0.1133	0.22	0.6482	1.39	0.2920
X_1_ X_3_	1	4.96	0.0765	17.23	0.0032	4.50	0.0873
X_2_ X_3_	1	0.0057	0.9428	68.91	<0.0001	52.49	0.0008
X_1_^2^	1	8.96	0.0303	-	-	1.28	0.3097
X_2_^2^	1	0.18	0.6907	-	-	1.34	0.2985
X_3_^2^	1	26.60	0.0036	-	-	23.94	0.0045
Lack of Fit	3	2.72	0.2806	0.91	0.6079	4.85	0.1758
*R^2^*		0.9383		0.9167		0.9519	
*Adjust R^2^*		0.8272		0.8541		0.8652	

DF = degree of freedom, 95% significant level. X_1_ = CG concentration (g/L), X_2_ = initial pH, X_3_ = Shaking speed (rpm).

**Table 5 microorganisms-08-00453-t005:** Total reducing sugars consumption, glycerol consumption, cell mass yield, and lipid yield by *R. fluvialis* DMKU-SP314.

Stage (Cultivation Time)	TRS_C_ (% of Initial TRS)	CG_C_ (% of Initial CG)	Cell Mass (g/L)	Lipid Concentration (g/L)	Y_x_ (g/g)	Y_L_ (g/g)
1st stage (48 h)	69.54	-	10.61 ± 0.22	0.65 ± 0.09	0.35	0.02
2nd stage (192 h)	-	76.43	11.09 ± 0.23	15.20 ± 0.54	0.17	0.23

TRS_C_ = Percentage of total reducing sugar consumption (g/L), CG_C_ = Percentage of crude glycerol consumption (g/L).

**Table 6 microorganisms-08-00453-t006:** Comparative study of lipid productivity by oleaginous yeasts cultivated in various substrates using two-stage cultivation.

Yeast	Carbon Sources	Cell Mass (g/L)	Lipid Concentration (g/L)	Lipid Content (% of Dry Cell Mass)	Lipid Productivity (g/L/h)	Culture Mode (Volume)	References
*Rhodosporidiobolus fluvialis* DMKU-SP314	sugarcane top hydrolysate and crude glycerol	21.7	15.8	73.0	0.066	Two-stage batch flasks (100 mL)	This study
*Rhodosporidiobolus fluvialis* DMKU-SP314	sugarcane top hydrolysate and crude glycerol	38.5	23.6	61.9	0.098	Two-stage fed-batch bioreactor (3 L)	[21]
*Cryptococcus curvatus* ATCC 20509	crude glycerol	32.9	17.4	53.0	0.060	Two-stage fed-batch bioreactor (1.5 L)	[31]
*Cryptococcus curvatus* MUCL 29819	glucose and acetate	4.0	1.0	47.3	0.011	Two-stage batch flasks (250 mL)	[50]
*Rhodosporidiobolus fluvialis* DMKU-RK253	crude glycerol	22.4	16.0	70.0	0.074	Two-stage batch flasks (100 mL)	[23]
*Rhodotorula glutinis* TISTR 5159	palm oil mill effluent	8.8	4.6	51.9	0.064	Two-stage semi-continuous bioreactor (1 L)	[51]
*Rhodosporidium toruloides* AS 2.1389	glucose and acetate	6.8	2.1	50.1	0.011	Two-stage batch flasks (50 mL)	[52]
*Rhodosporidium toruloides* Y4	glucose and glycerol	20.3	8.6	42.5	0.051	Two-stage batch flasks (50 mL)	[34]
*Trichosporonoides spathulata* JU4–57	crude glycerol	13.8	7.8	56.4	0.065	Two-stage fed-batch bioreactor (2 L)	[33]
*Yarrowia lipolytica* TISTR 5151	serum latex and glycerol	7–8	3.4	44.5	0.024	Two-stage batch flasks (50 mL)	[32]

**Table 7 microorganisms-08-00453-t007:** The fatty acids composition (%) produced by *R. fluvialis* DMKU-SP314 when grown on STH and its comparison with other plant oils.

Source	C14:0	C16:0	C18:0	C18:1	C18:2	C18:3
*Rhodosporidiobolus fluvialis* DMKU-SP314	1.7	31.0	5.4	36.7	20.7	4.4
Canola	-	4.5	2.0	60.4	21.2	9.4
Castor	-	1.4	1.1	3.4	4.8	0.6
Olive	0.1	11.3	2.8	74.5	9.8	0.5
Palm	1.1	42.3	4.3	40.9	10.1	0.3
Peanut	-	10.3	2.8	49.6	31.5	0.6
Rapeseed	-	4.1	1.5	62.3	20.6	8.7
Soybean	0.1	11.5	4.1	23.5	53.3	6.8
Sunflower	-	6.4	3.9	20.9	67.6	0.2

C14:0; myristic acid, C16:0; palmitic acid, C18:0; stearic acid, C18:1; oleic acid, C18:2; linoleic acid and C18:3; α-linolenic acid.

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
