# Peer review of "Lipid Production from Sugarcane Top Hydrolysate and Crude Glycerol with Rhodosporidiobolus fluvialis Using a Two-Stage Batch-Cultivation Strategy with Separate Optimization of Each Stage"

_microorganisms, 2020, doi:10.3390/microorganisms8030453_

Round 1

Reviewer 1 Report

Lipid Production from Sugarcane Top Hydrolysate and Crude Glycerol by Rhodosporidiobolus fluvialis 3 using a Two-stage Cultivation Strategy

This paper deals with the optimization of the production of lipids by Rhodosporidiobolus fluvialis DMKU_SP314 from Sugarcane Top Hydrolysate and Crude Glycerol by a two-stage fermentation. The optimization is interestingly made either through one-factor-at-a-time or with PBD and BBD, according to different variables tested: nitrogen sources, C/N, temperature, shaking speed or glycerol concentration.

Main comment : the text is sometimes hard to read as many grammatical errors occur, some commas are misplaced and some sentences are too long and/or not clear. The text must be checked  for paragraphs difficult to understand because of unclear sentences or misspelled word/ expressions.

Title:  When reading the title and the abstract the topic seems similar to the previous article “Feeding Strategies of Two-Stage Fed-Batch Cultivation Processes for Microbial Lipid Production from Sugarcane Top Hydrolysate and Crude Glycerol by the Oleaginous Red Yeast Rhodosporidiobolus fluvialis” (Rujiralai Poontawee and Savitree Limtong in  Microorganisms 2020, 8, 151)

The title should then be modified. The words “optimization by BBD and one factor at a time of lipids production….” or another formulation may appear.

It may be clearly announced (in introduction/ abstract) that the purpose is to implement the previous study and improve each step of the process using experimental design.

Similarly, in a “Conclusion part”, a comparison should be made between the results (concentrations/ yields) obtained in this 2 studies.

Other comments:

L28 and all along the text, figures and tables :

There are 2 variables : “Lipids concentration” and “lipid content” calculated in two different ways with 2 different units (mg/L and %): please use the same terminology (especially for “lipids concentration”) all along the text. Avoid changing in “lipid, lipids, lipid production, lipid concentration…”. I suggest to keep “lipids concentration” in the whole text/ figures/tables, except if talking about the generality : “lipid production”.

Moreover make the paragraph “2.4 analytical methods” clearer, to better explain what is the distinction between both variables.

L70-71: ‘The composition of ST is including 42.7% cellulose, 25.5% hemicellulose and 5.8% lignin content of the dry solid sugarcane top » : add a reference for this statement.

L78: “glycerol (GC) » rather CG?  as “GC” is glycerol consumption in Table 5?

L105: “dried under the sun. » Number of days?

L105: “passed through a 40-60 mesh size screen” : add SI unit for mesh size? mm? µm?

L146: “2.3.1.2. Screening of significant factors”  give details about the factors selected directly in the title or in the first sentence of the paragraph.

Which levels were selected for these factors? (to be mentioned)

L153: “2.3.1.3. Optimization by one-factor-at-a-time and Box-Behnken design”: add in the title the factors selected for these experiments Or add subtitles for each set of experiments with the factors selected.

L185: “Three variables, namely CG concentration “ Explain why CG concentration may be tested again whereas it has already been done in the previous work (Pontawee 2020).

L205: “Lipid content was expressed as the percentage of gram lipid concentration per gram dry cell mass (%of dry cell mass).” Do you mean: lipid concentration? or lipid mass?

L20-212: “To enhance lipid production from two low cost feedstocks i.e. STH and CG from a biodiesel production plant by the selected oleaginous strain, R. fluvialis DMKU-SP314, and separate optimization of each stage in the two-stage cultivation was used”: sentence not clear.

L217-219: write rather “To find the most appropriate additional nitrogen source for cell mass production in the first-217 stage, inorganic nitrogen compounds i.e. (NH4)2SO4 OR NH4Cl, and organic nitrogen compounds i.e. 218 CSL…”

L225 : “Figure 1. Cell mass (■), lipid content (■) and lipid (■) by R. fluvialis DMKU-SP314 cultivated” : colors are not visible in the caption.

Replace “Lipid” by “Lipids concentration”

L242-243: “Eight variables were analyzed with regard to their effects on cell mass production at the end of 242 first-stage using PBD” : mention here the 8 variables screened.

L248: “Generally, variables with higher t value and lesser p value…” : t or T value (as in table 1)?

L271 : why was the shaking speed set at 160-180-200 rpm?

L329 : add a subtitle : as an example : -Shift time improvement

L341 : add a subtitle : as an example : -TRS concentration

L344 : “approximately 40 g/L (only TRS in STH)” change as : “… (only natural TRS in STH)”

L352-358: “Figure 2. Time-course of cell mass (●), lipid content (▲) and lipid (■) by R. fluvialis DMKU-352 SP314 cultivated in STH medium consisting of 70 g/L TRS, 2.58 g/L SBP, 0.94 g/L (NH4)2SO4, 353 0.39 g/L KH2PO4, 2.5 g/L MgSO4•7H2O, with an initial pH of 6, a temperature of 28 °C and 354 a shaking speed of 200 rpm in the first stage and in the second stage, the cells were re-355 suspended in 70 g/L crude glycerol solution with a pH of 5.5 and incubated at 150 rpm and 356 28 °C at different shift times (a) 48 h (b) 72 h (c) 96 h and (d) 120 h. Data are presented as 357 mean value ± standard deviation.”

Make the figure caption clearer .

Same capital letters may appear in the figure caption as on the graph :A,B,C,D

L361-363 : (table 3) add the meaning of “TRS” in the table’s foot notes.

I think TRSc may be calculated in % (rather than g/L)

L386-391: Fiigure 3 caption may be clearer.

“Supplemented with SBF”? or rather SBP?

L395-400: Sentences too long, not clear.

L412-416 : as Y1, Y2, Y3 are mentioned in table 4 the equations may be written as :

Cell mass production

Y1=

Lipids concentration

Y2=

Lipid content

Y3=

L423-434 : « (R2) for cell mass, lipid production and lipid content were 423 relatively high at 0.94, 0.92, and 0.95”: are you sure it is high as R2. What is the limit?

L460-462: I think TRSc and Gc may be calculated in % (rather than g/L)

L478: Why the table 6 do not include the results from the previous study (Poontawee 2020)?

L483-484 :” The fatty acid compositions of lipids produced by R. fluvialis DMKU-SP314 were determined by GC after a transesterification step.”

Rather write “The fatty acid compositions of lipids produced by R. fluvialis DMKU-SP314 in this study was determined by gaz chromatography after a transesterification step.”

L486: “Similar fatty acid compositions were reported for R. fluvialis DMKU-RK253 grown on CG [19]” To be compared with the results from the previous work (Poontawee 2020)?

L490: rather write :“The fatty acid composition of lipid produced by this yeast species is qualitatively similar to that of the common plant oils used in biodiesel production”

A conclusion would be appreciated.

Author Response

We really appreciate all the comments and suggestions that have made much improvement of our manuscript.

Reviewer 1

Lipid Production from Sugarcane Top Hydrolysate and Crude Glycerol by Rhodosporidiobolus fluvialis 3 using a Two-stage Cultivation Strategy

This paper deals with the optimization of the production of lipids by Rhodosporidiobolus fluvialisDMKU_SP314 from Sugarcane Top Hydrolysate and Crude Glycerol by a two-stage fermentation. The optimization is interestingly made either through one-factor-at-a-time or with PBD and BBD, according to different variables tested: nitrogen sources, C/N, temperature, shaking speed or glycerol concentration.

Main comment: the text is sometimes hard to read as many grammatical errors occur, some commas are misplaced and some sentences are too long and/or not clear. The text must be checked for paragraphs difficult to understand because of unclear sentences or misspelled word/ expressions.

Response: Thank you very much, The revised MS was the English language edited again by a Native English speaker, who is not the one that edited the original MS.

Title:  When reading the title and the abstract the topic seems similar to the previous article “Feeding Strategies of Two-Stage Fed-Batch Cultivation Processes for Microbial Lipid Production from Sugarcane Top Hydrolysate and Crude Glycerol by the Oleaginous Red Yeast Rhodosporidiobolus fluvialis” (Rujiralai Poontawee and Savitree Limtong in Microorganisms 2020, 8, 151)

The title should then be modified. The words “optimization by BBD and one factor at a time of lipids production….” or another formulation may appear.

Response: The title was changed to “Lipid Production from Sugarcane Top Hydrolysate and Crude Glycerol with Rhodosporidiobolus fluvialis using a Two-Stage Batch Cultivation Strategy with a Separate Optimization of Each Stage”

            This is because in the previous paper we also used BBD and one factor at a time. The difference is in the present study we used two-stage batch cultivation and optimization of the first-step (for high cell mass production) and the second step was performed separately but in the previous study fed-batch was used and no separate optimization of the first and second step was done.

It may be clearly announced (in introduction/ abstract) that the purpose is to implement the previous study and improve each step of the process using experimental design.

Response: In this study, we did not implement the previous study by using experimental design. As mention above in the previous paper, we also used BBD and one factor at a time for the optimization of lipid production by two-stage fed-batch cultivation. The difference is in this the present study we used two-stage batch cultivation and optimization of the first-step (for high cell mass production) and the second-step (for high lipid production) were performed separately but in the previous study fed-batch was used and no separate optimization of the first and second step was done.

However, to show that the previous work and this work are different we added “In a recent study, R. fluvialis DMKU-SP314 was used for lipid production from sugarcane top hydrolysate and crude glycerol in a two stage fed‐batch cultivation processes [21].“ in Line 95-97

Similarly, in a “Conclusion part”, a comparison should be made between the results (concentrations/ yields) obtained in this 2 studies.

Response:  The comparison between the results (concentrations/ yields) obtained in this 2 studies was added in table 6 and discussed as “Comparing this result with the same yeast strain and substrate revealed that using two-stage batch cultivation could enhance lipid content better than two-stage fed-batch cultivation that was used in the previous study [21].” (Line 477)

Other comments:

L28 and all along the text, figures and tables :

There are 2 variables : “Lipids concentration” and “lipid content” calculated in two different ways with 2 different units (mg/L and %): please use the same terminology (especially for “lipids concentration”) all along the text. Avoid changing in “lipid, lipids, lipid production, lipid concentration…”. I suggest to keep “lipids concentration” in the whole text/ figures/tables, except if talking about the generality : “lipid production”.

Response:  All were changed to “Lipid concentration” as your suggestions.

Moreover make the paragraph “2.4 analytical methods” clearer, to better explain what is the distinction between both variables.

Response:  the sentences “Lipid concentration was expressed as grams of the lipid (the sum of fatty-acid concentration) per milliliter of the culture broth (g/L). Lipid content was expressed as the percentage of grams of lipid concentration per grams of dry cell mass (% of dry cell mass).” was added to line 219-221

L70-71: ‘The composition of ST is including 42.7% cellulose, 25.5% hemicellulose and 5.8% lignin content of the dry solid sugarcane top » : add a reference for this statement.

Response:  The reference was added “The composition of dry ST including 42.7% cellulose, 25.5% hemicellulose and 5.8% lignin [29].”-line 84

L78: “glycerol (GC) » rather CG?  as “GC” is glycerol consumption in Table 5?

Response:  CG refers to Crude glycerol (Line 86), therefore GC in table 5 was changed to CGc (Crude glycerol consumption).

L105: “dried under the sun. » Number of days?

Response:  We dried the ST for 2 days. Therefore for 2 days was added at the end of the sentence “For the preparation of STH, ST harvested from the field was washed with tap water to remove dirt and then dried under the sun for 2 days”(Line 120)

L105: “passed through a 40-60 mesh size screen” : add SI unit for mesh size? mm? µm?

Response:  The SI unit (Sieve size 0.389 -0.231 mm) was added to be “Dried ST was milled and passed through a 40-60 mesh sized screen (Sieve size 0.389 -0.231 mm).”(Line 121)

L146: “Screening of significant factors”  give details about the factors selected directly in the title or in the first sentence of the paragraph.

Response:  The following sentence was moved to the first sentence of paragraph “In this study, eight factors consisting of medium component parameters (X1; concentration of TRS, X2; additional nitrogen sources, X3; (NH4)2SO4, X4; KH2PO4, and X5; MgSO4•7H2O) and physical parameters (X6; initial pH, X7; temperature and X8; shaking speed) were selected.” (Line 156-158)

Which levels were selected for these factors? (to be mentioned)

Response:  The factors with a 90% confidence level were identified as significant factors. Therefore, the sentence “The significant factors for cell mass and lipid production with levels of confidence higher than 90%, and the levels that gave the highest concentration of cell mass and lipid were selected for further studies” was added. (Line 160)

L153: “2.3.1.3. Optimization by one-factor-at-a-time and Box-Behnken design”: add in the title the factors selected for these experiments Or add subtitles for each set of experiments with the factors selected.

Response: 2.3.1.3. was divided into 3 titles

2.3.1.3. Optimization temperature by a one-factor-at-a-time method

2.3.1.4 Optimization of the concentration of (NH4)2SO4 and KH2PO4, and shaking speed by Box-Behnken design

2.3.1.5. Shift time improvement

2.3.1.6. Initial TRS concentration

L185: “Three variables, namely CG concentration “Explain why CG concentration may be tested again whereas it has already been done in the previous work (Pontawee 2020).

Response: In this study, the three variables (GC, initial pH and shaking speed) were tested to investigate the interaction effects of each variable on lipid production by using experimental design. The sentence was changed to “Three factors, namely CG concentration, initial pH and shaking speed, were used to investigate the interactive effects of each variable on lipid production by using the experimental design.” (Line 199)

L205: “Lipid content was expressed as the percentage of gram lipid concentration per gram dry cell mass (%of dry cell mass).” Do you mean: lipid concentration? or lipid mass?

Response: It is lipid content.

L20-212: “To enhance lipid production from two low cost feedstocks i.e. STH and CG from a biodiesel production plant by the selected oleaginous strain, R. fluvialis DMKU-SP314, and separate optimization of each stage in the two-stage cultivation was used”: sentence not clear.

Response:  The sentence was changed to “To enhance lipid production from two low-cost substrates i.e. STH and CG by the selected oleaginous strain, R. fluvialis DMKU-SP314, a two-stage batch cultivation was used. Separate optimizations of cell mass production in the first stage and lipid production in the second stage were performed.” (Line 225)

L217-219: write rather “To find the most appropriate additional nitrogen source for cell mass production in the first-217 stage, inorganic nitrogen compounds i.e. (NH4)2SO4 OR NH4Cl, and organic nitrogen compounds i.e. 218 CSL…”

Response:  The sentence was changed to “To find the most appropriate nitrogen source for cell mass production, inorganic nitrogen compounds i.e. (NH4)2SO4 and NH4Cl, and organic nitrogen compounds i.e. CSL, C5H8NNaO4, SBP, CO(NH2)2 and YE were individually tested at the same nitrogen concentration. At the end of the first stage, the highest cell masses of 9.62, 9.60, 9.08 and 8.74 g/L were obtained in the STH medium supplemented with CO(NH2)2, CSL, SBP, and C5H8NNaO4, respectively (Figure 1).”(Line 233)

L225 : “Figure 1. Cell mass (■), lipid content (■) and lipid (■) by R. fluvialis DMKU-SP314 cultivated” : colors are not visible in the caption.

Response:  Sorry for my mistake, the colors were added.

Replace “Lipid” by “Lipids concentration”

Response:  Replaced “Lipid” by “Lipids concentration”

L242-243: “Eight variables were analyzed with regard to their effects on cell mass production at the end of 242 first-stage using PBD” : mention here the 8 variables screened.

Response: Eight variables including; concentration of TRS, additional nitrogen source, (NH4)2SO4, KH2PO4, MgSO4•7H2O, initial pH, temperature and shaking speed were mention in line 256 as “Eight factors (including the concentration of TRS, the additional nitrogen source, (NH4)2SO4, KH2PO4, MgSO4•7H2O, the initial pH, temperature and shaking speed) were analyzed”.

L248: “Generally, variables with higher t value and lesser p value…” : t or T value (as in table 1)?

Response:  t value was changed to T value

L271 : why was the shaking speed set at 160-180-200 rpm?

Response:  To investigate the optimum condition of shaking speed. The speed set was from the result of PBD.

L329 : add a subtitle : as an example : -Shift time improvement

Response:  Shift time improvement was added as a subtitle.

L341 : add a subtitle : as an example : -TRS concentration

Response:  TRS concentration was added as a subtitle.

L344 : “approximately 40 g/L (only TRS in STH)” change as : “… (only natural TRS in STH)”

Response: (only TRS in STH) was changed to (only natural TRS in STH).

L352-358: “Figure 2. Time-course of cell mass (●), lipid content (▲) and lipid (■) by R. fluvialis DMKU-352 SP314 cultivated in STH medium consisting of 70 g/L TRS, 2.58 g/L SBP, 0.94 g/L (NH4)2SO4, 353 0.39 g/L KH2PO4, 2.5 g/L MgSO4•7H2O, with an initial pH of 6, a temperature of 28 °C and 354 a shaking speed of 200 rpm in the first stage and in the second stage, the cells were re-355 suspended in 70 g/L crude glycerol solution with a pH of 5.5 and incubated at 150 rpm and 356 28 °C at different shift times (a) 48 h (b) 72 h (c) 96 h and (d) 120 h. Data are presented as 357 mean value ± standard deviation.”

Make the figure caption clearer.

Same capital letters may appear in the figure caption as on the graph: A, B, C, D

Response:  Sorry for my mistake, the colors were added, and the same capital letters were edited.

L361-363 : (table 3) add the meaning of “TRS” in the table’s foot notes.

Response:  The meaning of “TRS” was added in the table’s footnotes as “a TRS = initial TRS concentration (g/L).”

I think TRSc may be calculated in % (rather than g/L)

Response:  the TRSc was changed to calculate in % as your suggestion.

L386-391: Fiigure 3 caption may be clearer.

Response:  The caption of figure 3 was changed to “ The effect of temperature on (A) cell mass production (g/L) when cultivated in an STH medium supplemented with SBP at 120 h in a temperature gradient incubator (15.0–35.5 °C). The effect of temperature on (B) lipid concentration (g/L) when cultivated in a crude glycerol solution for 196 h (after cultivation in the first stage for 48 h) in a temperature gradient incubator (17.0–37.8 °C). Data were presented as mean value ± standard deviation.”

“Supplemented with SBF”? or rather SBP?

Response:  Sorry for my mistake, SBF was changed to SBP

L395-400: Sentences too long, not clear.

Response:  The sentence was changed to “When the R. fluvialis DMKU-SP314 was cultivated under optimum conditions in the first stage, and within various conditions during the second stage according to the BBD, 17 experimental runs were conducted. The results show cell mass concentrations ranging between 20.99-26.07 g/L, lipid concentrations between 10.84-14.16 g/L and lipid content between 41.64-57.39% of dry cell mass The ANOVA for the response surface quadratic model for cell mass production, lipid production, and lipid content are presented below in Table 4.”(Line 412-417).

L412-416 : as Y1, Y2, Y3 are mentioned in table 4 the equations may be written as :

Cell mass production

Y1=

Lipids concentration

Y2=

Lipid content

Y3=

Response:  The regression equations were changed as your suggestions.

L423-434 : « (R2) for cell mass, lipid production and lipid content were 423 relatively high at 0.94, 0.92, and 0.95”: are you sure it is high as R2. What is the limit?

Response:  The sentences were changed to “This statement is supported by the fact that the values of the correlation coefficient (R2) for cell mass production, lipid concentration, and lipid content were 0.94, 0.92, and 0.95, respectively, which in turn suggests that most of the errors/variation in the model can be explained [47].”(Line 429)

L460-462: I think TRSc and Gc may be calculated in % (rather than g/L)

Response:  TRSc and Gc was changed to calculated in % (Table 5)

L478: Why the table 6 do not include the results from the previous study (Poontawee 2020)?

Response:  The previous study (Poontawee 2020) was added in Table 6

L483-484 :” The fatty acid compositions of lipids produced by R. fluvialis DMKU-SP314 were determined by GC after a transesterification step.”

Rather write “The fatty acid compositions of lipids produced by R. fluvialis DMKU-SP314 in this study was determined by gaz chromatography after a transesterification step.”

Response:  The sentence was changed as your suggestion.

L486: “Similar fatty acid compositions were reported for R. fluvialis DMKU-RK253 grown on CG [19]” To be compared with the results from the previous work (Poontawee 2020)?

Response:  the previous work was compared. Therefore the sentence “Similar fatty acid compositions were reported for R. fluvialis DMKU-SP314 in previous study [21], for R. fluvialis DMKU-RK253 grown on CG [23], and R. toruloides grown on corn stover hydrolysate [53].” was added. (Line 496)

L490: rather write :“The fatty acid composition of lipid produced by this yeast species is qualitatively similar to that of the common plant oils used in biodiesel production”

Response: The sentence was changed as your suggestion.

A conclusion would be appreciated.

Response:  The conclusion was added as your suggestion as “This study demonstrated that STH is a suitable substrate for cell mass production by R. fluvialis DMKU–SP314. The use of a two stage batch cultivation led to high cell density and an increase in a high quantity of lipids. Moreover, CG was found to be an effective carbon source for lipid production in the second stage of cultivation with R. fluvialis DMKU–SP314. The lipid volume produced by this yeast species exhibited a fatty acid composition similar to that of major plant oils commonly used in biodiesel production currently. Therefore, R. fluvialis DMKU–SP314 has a promising ability to convert agricultural residues and hydrolysates into microbial lipid with no additional nitrogen source needed in the lipid accumulation stage in order to produce lipid content viable as renewable biodiesel fuel.” (Line 507-516)

Reviewer 2 Report

The topic of this manuscript is very interested and I am glad that someone is doing this research.

The main criticism is that the English grammar has to be improved. And, there are multiple occurrences where (obviously) words are missing.

Abstract:

Remove the word 'the' from the beginning of the second sentence.

Ln 18: insert the word "production" between feedstock and up & insert the word "an" between to and industrial.

Ln 19: remove the word "the"

Ln 22: insert the word "a" between in and crude

Ln 27: "second stage" is singular, please chnage

Introduction

In your first sentence you make a comment that biodiesel is an environmentally friendly fuel. I suppose that I know what you are trying to say - it is a renewable carbon source. However, any form of diesel cannot be considered to be an environmentally friendly fuel.

In your second sentence you make a comment that biodiesel is used in combination with petrol. That is not really true - diesel and petrol are quite different fuels and are not interchangeable.

Ln 35: change the word "it" to "biodiesel"

Ln 35: in your stated examples of sources of feedstocks for biodiesel you are only mentioning plant-based products. What about animals based lipid feedstocks?

Ln 38: which over the words "energy" and "current" so that it reads "current energy". Further to this point, Is there a current energy crisis?

Ln 46: you state that the growth rate decelerates when N is exhausted. But, isn't that true for all microorganisms?

Ln 78: your abbreviation is the wrong way around - it should read "CG"

Ln 105: please complete the sentence after the word "to"

Above are a series of (on an individual basis) quite minor shortcoming. However, there are far too many of these. You will have to go through this manuscript with a "very fine toothcomb" and remove these relatively minor errors before a proper review of your scientific paper can be reviewed properly.

Author Response

We really appreciate all comments and suggestions that have made much improvement in our manuscript.

Reviewer 2

The topic of this manuscript is very interested and I am glad that someone is doing this research.

The main criticism is that the English grammar has to be improved. And, there are multiple occurrences where (obviously) words are missing.

Response: Thank you very much, The revised MS was the English language edited again by a Native English speaker, who is not the one that edited the original MS.

Abstract:

Remove the word 'the' from the beginning of the second sentence.

Response: removed

Ln 18: insert the word "production" between feedstock and up & insert the word "an" between to and industrial.

Response: the word "production" was inserted.

Ln 19: remove the word "the"

Response: We removed “the” and combined these two sentences “This study aimed to enhance lipid production from low-cost substrates by two-stage cultivation process using Rhodosporidiobolus fluvialis DMKU-SP314 and the two low-cost substrates namely sugarcane top and biodiesel-derived crude glycerol. In the first-stage, sugarcane top hydrolysate was used for cell propagation, and in the second stage, cells were suspended in crude glycerol solution for lipid production.”  to “This study aimed to enhance lipid production from low-cost substrates namely sugarcane top and biodiesel-derived crude glycerol by two-stage cultivation process using Rhodosporidiobolus fluvialis DMKU-SP314.”

Ln 22: insert the word "a" between in and crude

Response: inserted the word "a" between in and crude

Ln 27: "second stage" is singular, please change

Response: changed to "second stage"

Introduction

In your first sentence you make a comment that biodiesel is an environmentally friendly fuel. I suppose that I know what you are trying to say - it is a renewable carbon source. However, any form of diesel cannot be considered to be an environmentally friendly fuel.

Response: we deleted “an environmentally friendly” and added “as it is renewable and less toxic. The combustion of biodiesel is very similar to fossil fuel combustion, but it produces less harmful gases, such as sulfur oxide, sulfur dioxide, and sulfur trioxide.” Therefore, the first two sentences in the revised MS were “Biodiesel has largely been accepted as an alternative energy source to fossil fuel, as it is renewable and less toxic. The combustion of biodiesel is very similar to fossil fuel combustion, but it produces less harmful gases, such as sulfur oxide, sulfur dioxide, and sulfur trioxide [1].” (Line 34)

In your second sentence you make a comment that biodiesel is used in combination with petrol. That is not really true - diesel and petrol are quite different fuels and are not interchangeable.

Response: We changed this sentence to “Currently, biodiesel is being used worldwide as the main source of fuel for diesel engine machinery without the need for major modifications to the mechanics of these machines [2]. (Line 37)

Ln 35: change the word "it" to "biodiesel"

Response: changed the word "it" to "biodiesel"

Ln 35: in your stated examples of sources of feedstocks for biodiesel you are only mentioning plant-based products. What about animals based lipid feedstocks?

Ln 38: which over the words "energy" and "current" so that it reads "current energy". Further to this point, Is there a current energy crisis?

Response: From these two comments, we added information of animal-based lipid feedstocks. Also, at present there no energy crisis so rewrote this statement as follows.

 “Biodiesel can be divided into three generations based on the feedstock which generates the fuel. First-generation biodiesel is produced from edible plant oils, such as palm oil, soybean oil, and coconut oil, and while second-generation biodiesel is produced from nonedible plant oils, such as jatropha, animal fats and waste oils [3]. The recent current generation of biodiesel is derived from microbial lipids. Using recovered animal fats and frying oils of the second generation as feedstock for biodiesel can efficiently reduce the price of the fuel; however, the amount of these fats and oils is limited on an industrial scale and cannot meet the increasing needs of biodiesel production [4]. Recently, microbial lipids that are produced by oleaginous microorganisms have been getting increased attention as oil feedstock [5]. In terms of obtainability and sustainability, lipids from oleaginous microorganisms are recognized as opportunity feedstocks for biodiesel production and provide a promising route to food and energy security in the current energy crisis [6].” (Line 39)

Ln 46: you state that the growth rate decelerates when N is exhausted. But, isn't that true for all microorganisms?

Response: Corrected to “In oleaginous yeasts,”

Ln 78: your abbreviation is the wrong way around - it should read "CG"

Response:  changed from “GC” to "CG" throughout the revised MS

Ln 105: please complete the sentence after the word "to"

Response: added “remove dirt.” after “to” (Line 120)

Above are a series of (on an individual basis) quite minor shortcoming. However, there are far too many of these. You will have to go through this manuscript with a "very fine toothcomb" and remove these relatively minor errors before a proper review of your scientific paper can be reviewed properly.

Response: Thank you for your kind comments. We carefully checked throughout the MS and revised all errors.

Reviewer 3 Report

This study aimed to enhance lipid production from low-cost substrates by two-stage cultivation process using Rhodosporidiobolus fluvialis DMKU-SP314 and the two low-cost substrates namely sugarcane top and biodiesel-derived crude glycerol. In the first-stage, sugarcane top hydrolysate was used for cell propagation, and in the second-stage, cells were suspended in crude glycerol solution for lipid production. Optimization for high cell mass production in the first-stage and for high lipid production in the second-stage were performed separately by using a one-factor-at-a-time together with response surface methodologies. Under the optimum nutrients and conditions of the first-stage and second-stages, a high lipid quantity,a high cell mass and a high lipid content of dry cell mass were obtained. This study was well designed and reasonable explained. This manuscript can be accepted for publication after minor revision. Other comments were as followed.

1) Materials and methods part is too long, many of the mehtod is usually and commonly used method. So the author should improve and revise this part to a concise form.

2) The author used one factor at a time. How about multi-factor at a time? why not use mutli-factor? please explain or discusse it.

3)Under the optimum condition, the target product was obtained. The author should compare your production to the published references, to show your good results.

4) After you get the final result, what is the novelty of this study? please indicate it in the text.

5) What is the aspect of this study used for? Please provide some usage fields.

Author Response

We really appreciate all comments and suggestions that have made much improvement in our manuscript.

Reviewer 3

Comments and Suggestions for Authors

This study aimed to enhance lipid production from low-cost substrates by two-stage cultivation process using Rhodosporidiobolus fluvialis DMKU-SP314 and the two low-cost substrates namely sugarcane top and biodiesel-derived crude glycerol. In the first-stage, sugarcane top hydrolysate was used for cell propagation, and in the second-stage, cells were suspended in crude glycerol solution for lipid production. Optimization for high cell mass production in the first-stage and for high lipid production in the second-stage were performed separately by using a one-factor-at-a-time together with response surface methodologies. Under the optimum nutrients and conditions of the first-stage and second-stages, a high lipid quantity, high cell mass and a high lipid content of dry cell mass were obtained. This study was well designed and reasonable explained. This manuscript can be accepted for publication after minor revision. Other comments were as followed.

1) Materials and methods part is too long, many of the mehtod is usually and commonly used method. So the author should improve and revise this part to a concise form.

Response: Thank you very much for your kind suggestion. We revised the materials and methods part.

2) The author used one factor at a time. How about multi-factor at a time? why not use mutli-factor? please explain or discusse it.

Response: We used one-factor-at-a-time and multi-factor at a time by response surface methodology using Box-Behnken design. Single was used to investigate an accurate optimal temperature of yeast growth and lipid production by using the temperature gradient incubator. The multi-factors by RSM were used to obtain the optimum condition of multifactor using a shaking flask incubator.

To make clearer we divided 3.1 Optimization of cell mass production from sugarcane top hydrolysate in the first-stage to subtitle as;

3.1.1. Effect of additional nitrogen source

3.1.2 Screening of significant factors

3.1.3. Optimization of temperature by a one-factor-at-a-time method

3.1.4. Optimization of the concentration of (NH4)2SO4, KH2PO4, and shaking speed by Box-Behnken design

We also added the sentence “To obtain an accurate optimal temperature yeast growth, temperature optimization was conducted separately using a temperature gradient incubator.” (Line 272).

3) Under the optimum condition, the target product was obtained. The author should compare your production to the published references, to show your good results.

Response: We added the flowing sentences in the revised MS.

“In the first stage, when STH was used as a feedstock, there was a cell mass yield of up to 0.35 g/g TRS with only a low lipid yield of 0.02 g/g. This supported STH as a suitable carbon source for cell growth. Then, high lipid yield was observed in the second stage using a CG solution without auxiliary nutrients. A higher lipid yield of 0.23 g/g and cell mass yield of 0.17 g/g glycerol was obtained. These results, therefore, demonstrate that the two-stage cultivation process, in which STH was used in the first stage and CG was used in the second stage, produced optimal cell growth and lipid production. This provided the highest lipid content (73% of dry cell mass) when compared with other investigations, as shown in Table 6 (Line 470-477).

4) After you get the final result, what is the novelty of this study? please indicate it in the text.

Response: We added the sentence “This demonstrated that STH is a suitable substrate for cell mass production with R. fluvialis DMKU–SP314. The use of a two-stage cultivation process led to high cell density and an increase in a high quantity of lipids. Moreover, CG was found to be an effective carbon source for lipid production in the second stage of cultivation with R. fluvialis DMKU–SP314.” (Line 486).

5) What is the aspect of this study used for? Please provide some usage fields.

Response: We added the sentence “Therefore, R. fluvialis DMKU–SP314 has a promising ability to convert agricultural residues and hydrolysates into microbial lipids with no additional nitrogen source needed in the lipid accumulation stage in order to produce lipid content viable as renewable biodiesel fuel. (Line 513).

Round 2

Reviewer 2 Report

The authors have made adequate improvements to their manuscript and I recommend acceptance of this paper.